# From carbohydrates to fat: Trends in food intake among Swedish nutrition students from 2002 to 2017

**Maria Bergström**[1]*, **Andreas Håkansson**[2], **Anna Blücher**[1], **Håkan S. Andersson**[1,3]*

**1** Department of Chemistry and Biomedical Sciences, Linnaeus University, Kalmar, Sweden, **2** Department of Food Technology, Engineering and Nutrition, Lund University, Lund, Sweden, **3** Department of Medical Biochemistry and Biophysics, Karolinska Institute, Solna, Sweden

* maria.bergstrom@lnu.se (MB); hakan.andersson@ki.se (HSA)

**Data Availability Statement:** All relevant data are within the manuscript and its Supporting Information files.

## Abstract

Earlier studies have implied a change in dietary habits of the Swedish population towards a low carbohydrate, high fat diet. Questions have been raised about the development in recent years and potential health effects. We have investigated the dietary intake of Swedish female students enrolled in a university nutrition course between 2002 and 2017. The students carried out self-reporting of all food and drink intake over one weekday and one weekend day. Intake of macronutrients (E%) and micronutrients were calculated for the whole period while statistical analysis was performed for changes between 2009 and 2017 (729 women). Results showed significant changes in carbohydrate intake (from 47.0 to 41.4 E%) and fat intake (from 31.7 to 37.5 E%). Carbohydrate intake was significantly lower than the Nordic Nutrition Recommendations (45–60 E%). However, daily fiber intake remains high (3.0 g/MJ) in a national context, and intake of vitamin D and folate appears to increase during the period. The results suggest that the observed national transition from carbohydrate to fat intake persists, and that it might be especially evident among individuals interested in food and nutrition. Considering the fiber and micronutrient intake, the change is not necessarily unfavorable for this particular group.

## Introduction

Studies of food intake in Sweden have noted a change in energy proportions from protein, fat and carbohydrates that began about 2005–2010 [1, 2]. A shift in energy intake from carbohydrates towards fat and protein seems to continue in Sweden, as well as other high-income countries, but there is a lack of recent studies to confirm this [3]. The Swedish National Food Agency has monitored the dietary habits of the Swedish adult population every 10 years and reports from 1988, 1997–1998 and 2010–2011 are available [4]. The latest survey included 1005 women and 792 men which were divided into four age groups; 18–30, 31–44, 45–64 and 65–80 years. It was concluded in this study that the energy contribution (E%) among women and men from protein (17–18 E%) was within the recommended range, but fat intake (35 E%) was deemed too high whereas carbohydrate intake (47 E%) was too low. The Swedish

**Funding:** This study received support from the Linnaeus University. The funders had no role in study design, data collection and analysis, decision to publish, or preparation of the manuscript.

**Competing interests:** The authors have declared that no competing interests exist.

recommended ranges were, at this time, 10–20 E% from protein, 25–35 E% from fat and 50–60 E% from carbohydrates (SNR 2005, [5]). Recommendations were however adjusted shortly after the survey, and using these new guidelines (NNR 2012, [6]), the obtained E% values were within the recommendations (10–20 E% from protein, 25–40 E% from fat and 45–60 E% from carbohydrates). The survey showed that all groups in the study had a fiber intake below the recommended level (25–35 g/day or 3 g/MJ) and that young women (18–30 years) had the lowest average intake; 17.3 g/day or 2.3 g/MJ. The intake of some micronutrients was identified to be lower than recommended: vitamin D, folate, iron and potassium; intake of sodium was higher than recommended. A trend towards a higher energy contribution from protein in combination with a lower energy contribution from fat was also reported. The energy contribution from carbohydrates were, however, reported to be fairly stable during the period (1988–2011) but short-term fluctuations and recent trends are not covered in these surveys. Such changes might be detected by the annual estimations of the food consumption in Sweden available from the statistics collected by the Swedish Board of Agriculture. These statistics reflect the volumes of food that are sold to Swedish consumers (directly or indirectly) but do not account for losses (purchased foods that are never consumed). A report based on the annual statistics from 1960 to 2013 [7] shows that the increase in protein consumption is part of a long-term change, which is accompanied by a relative decrease in both carbohydrate and fat consumption (expressed as E%). A change in this general trend was identified from 2011, where a decrease in E% from both carbohydrates and protein was seen in favor of fat. The authors explain this with impact from diets that are low in carbohydrates and high in fat (low-carb, high-fat diets), that got high attention in Scandinavia from 2005 and onwards [8–10].

A similar dietary change was identified in data from the Northern Sweden diet database (NSDD) that is part of the Västerbotten intervention program (VIP) and the Northern Sweden MONICA project, which are large ongoing independent cross-sectional health surveys that have been pursued on the population of northern Sweden since 1986. A recent study from this cohort [1], that included more than 120 000 food reports from both men and women between 1986 and 2010 (25 years), showed a decrease in E% from fat in the beginning of the period that continued to about 2005; thereafter the E% from fat increased and was actually higher than for 1986 in the end of the period (last year in the published material was 2010). The change in energy from carbohydrates showed an inverted pattern, and E% from carbohydrates was lower in 2010 than in 1986, and the authors concluded that this change coincides with low-carb, high-fat diets being recognized and increasingly popular. These findings were confirmed in a recent study based on 8354 women and 7641 men from the same cohort [2] who reported their food intake twice, ten years apart (the first report in 1996–2004 and a second in 2005–2014); carbohydrate intake among women decreased from 52.3 to 49.0 E% and fat intake increased from 32.7 to 34.6 E%. A similar trend was reported for the German population over the period 2005–2013; the German National Nutrition Survey (NEMONIT; [3]) included 1062 women and 778 men. In this cohort, the carbohydrate intake among women decreased from 48.9 E% to 45.8 E% during the study period, while fat intake increased from 33.8 E% to 36.8 E%. A similar shift from carbohydrates to fat were also seen among men in all studies.

It is not evident from these reports if the progress towards diets with a low-carbohydrate, high-fat profile has continued to present day. There are, however, reasons to believe that the observed transition in diets is more general than what has been recognized in published materials, because attitudes towards fat have changed not only in the general population [11], but also in the scientific community [12], which has affected official dietary recommendations. One example is the latest US guidelines from 2015 where the upper limit for fat intake has been removed [13].

Diet is an important risk factor concerning death and disability (DALY) as defined by the Global Burden of Disease (GBD) Study [14]. The top three risk factors for women in 2016 were high blood pressure, high body-mass index (BMI) and high fasting plasma glucose [14]. The increasing BMI of the population is of major concern in Sweden [15] as well as in most other countries (including both high and low-income countries). Recent reports [16, 17] from Sweden estimated the prevalence of overweight (BMI 25–29.9 kg/m$^2$) to be about 30% for women and about 45% among men. The prevalence of obesity (BMI $\geq$30 kg/m$^2$) in these studies was approximately 15% for both sexes. Furthermore, there are no convincing data to support a decrease or levelling off in these values [18]. The underlying explanation behind the continuous increase in BMI is complex but can in part be ascribed to changes with regard to socioeconomic and life-style factors but also to changes in food consumption patterns [11, 17, 19]. The association of overweight and obesity with chronic diseases *e.g.* type 2 diabetes, high blood pressure, cardiovascular disease, asthma and arthritis, is well-established [20]. Diet can influence the development of chronic diseases, either directly or indirectly through the development of overweight and obesity, and thus warrants close monitoring in the general population.

In the present study, we report food survey data from female university-level students partaking in a basic level nutrition course at the Linnaeus University in the southeast of Sweden from 2002 to 2017. The study aims to identify and analyze changes in dietary habits over time in this sample. As outlined above, other studies have implied that a change in energy intake from protein, fat and carbohydrates is ongoing in Sweden, and therefore food intake data from the period 2002–2017 is of special interest in order to study this trend in different subsets of the population.

## Materials and methods

### Study design and food intake measurements

The study is based on data collected in an annual course in human nutrition ('Food, Nutrition and Health'), at the Linnaeus University, Kalmar, Sweden, between 2002 and 2017. Data was collected as a series of independent cross-sectional surveys, which involved a total of 1882 students (1610 females and 272 males), Table 1. The course was given at campus 2002–2008 and transferred into a distance format in 2009. To follow the geographical distribution effects of this transition, the average distance in kilometers from the Linnaeus University course site in Kalmar (Sweden) to the residence of registered students as well as the geographical distribution of course students in relationship to the total population of the respective county [21] was calculated for all years, Fig 1.

Irrespective of campus or distant format, the course corresponded to 5 weeks of full-time studies, and the students conducted the recordings approximately two thirds into the course (April–May). Prior to participation, the students were given an introduction to nutrition assessments in general and diet recordings in particular, and they were subsequently instructed to record and report all foods and amounts thereof (preferably by direct weighing) consumed during at least one weekday (Monday-Thursday) and one weekend day (Friday-Sunday). The students then reported the data using commercial dietetic software (latest version of Dietist, Kost & Näringsdata, Bromma, Sweden). Parameters such as sex, age, weight and length were also self-reported by the participants. Food records were submitted to the course coordinator who blinded the data prior to passing them on for data analysis.

### Exclusion criteria and study groups

Full open diet recordings (weighed food records) were obtained from 1069 females and 212 males, which is a response rate of 66% among women and 78% among men, Table 1 and Fig 2.

**Table 1. Characteristics of the study group.**

| | 2002 | 2003 | 2004 | 2005 | 2006 | 2007 | 2008 | 2009 | 2010 | 2011 | 2012 | 2013 | 2014 | 2015 | 2016 | 2017 | Total |
|---|---|---|---|---|---|---|---|---|---|---|---|---|---|---|---|---|---|
| Admitted males | 5 | 21 | 15 | 10 | 9 | 8 | 13 | 22 | 26 | 30 | 14 | 30 | 16 | 17 | 16 | 20 | 272 |
| Reports males | 4 | 17 | 13 | 10 | 6 | 8 | 12 | 12 | 17 | 23 | 12 | 18 | 15 | 11 | 7 | 14 | 212 |
| Admitted females | 30 | 28 | 18 | 35 | 28 | 43 | 144 | 139 | 131 | 204 | 156 | 160 | 106 | 133 | 134 | 121 | 1610 |
| Reports females | 28 | 22 | 14 | 23 | 23 | 28 | 72 | 85 | 97 | 120 | 119 | 120 | 78 | 82 | 85 | 73 | 1069 |
| Included females | 22 | 21 | 14 | 18 | 15 | 21 | 52 | 72[1] | 78[1] | 110[1] | 107[1] | 96[1] | 65[1] | 73[1] | 71[1] | 57[1] | 892[1] |
| Age (years) mean±SD | 25.3± 6.6 | 28.1 ±9.4 | 27.4± 6.6 | 26.9 ±5.4 | 29.7 ±8.1 | 32.9 ±11.2 | 35.8 ±11.9 | 33.8 ±11.2 | 33.0 ±10.2 | 28.8 ±8.8 | 28.9 ±9.5 | 27.5 ±8.0 | 29.0 ±9.0 | 28.6 ±9.6 | 31.8 ±10.7 | 31.6 ±11.9 | - |
| Weight (kg) mean±SD | 65.7 ±13.3 | 61.5 ±6.6 | 58.4 ±4.6 | 62.6 ±7.0 | 64.1 ±10.2 | 63.4 ±8.1 | 63.9 ±11.5 | 63.9 ±9.4 | 64.4 ±9.8 | 64.5 ±9.5 | 62.7 ±9.0 | 64.3 ±9.9 | 65.5 ±10 | 64.8 ±10 | 64.7 10 | 64.2 ±9.2 | - |
| Length (cm) mean±SD | n.d | n.d | n.d | n.d | n.d | n.d | n.d | 167 5.5 | 168 ±6.1 | 169 ±5.4 | 168 ±5.8 | 169 ±6.7 | 167 ±6.4 | 169 ±5.9 | 169 ±6.2 | 169 ±5.8 | - |
| BMI (kg/cm³) mean±SD | n.d | n.d | n.d | n.d | n.d | n.d | n.d | 22.8 ±3.0 | 22.8 ±3.1 | 22.7 ±3.5 | 22.2 ±2.6 | 22.5 ±3.1 | 22.9 ±3.1 | 22.4 ±3.1 | 22.8 ±3.7 | 22.5 ±2.8 | - |

[1]Data from 2009–2017 was used in statistical evaluation of food categories (see Table 2) and included in total 729 females. n.d. = not determined.

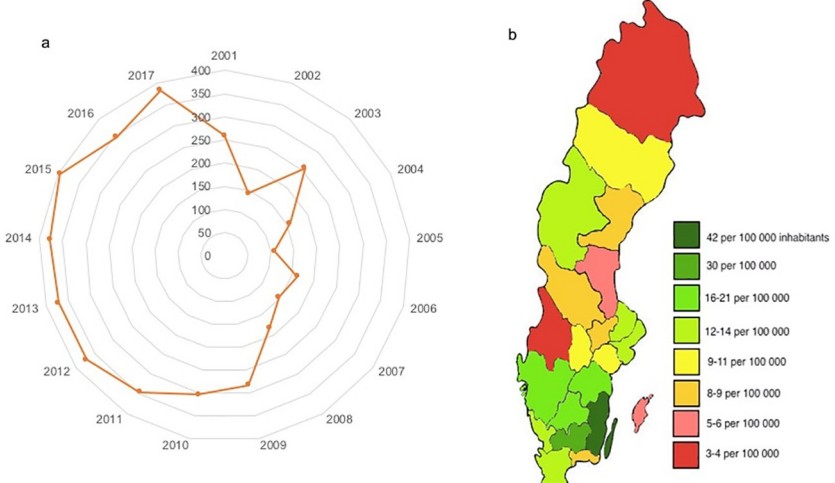

**Fig 1. Average distance and geographical distribution of participants.** a: Average distance in km from the Linnaeus university course site in Kalmar (Sweden) to the residence of registered students 2002–2017. b: Geographical distribution of course students registered 2009–2017 in relationship to the total population of the respective county. Population data from Statistics Sweden (21).

Due to the low number of male participants, only data from females were selected for further analysis. The respondents were assumed to have an average physical activity level (PAL) of 1.6, which corresponds to sedentary work and a somewhat active lifestyle outside of work, and a 95% Goldberg cutoff [22] was used to exclude individuals reporting unrealistic intake for any day. Almost all excluded values were due to underreporting, similar to what has been seen in previous investigations (4). After this filtering, 892 female respondents (83%) remained, Table 1 and Fig 2. Energy intake from protein, fat, carbohydrates, fiber and alcohol were calculated for this group for the years 2002–2017 in order to compare with official values from the

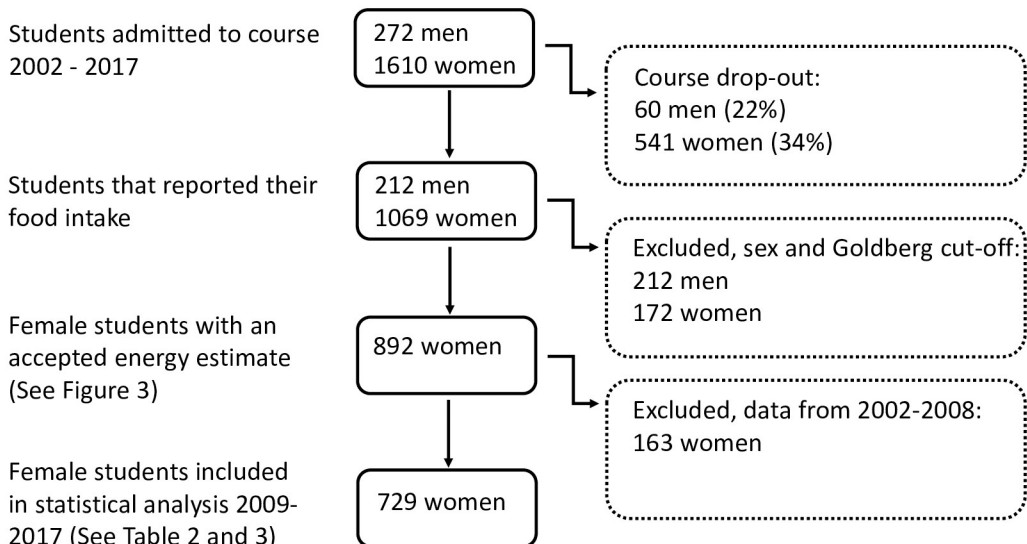

**Fig 2. Number of participants in the study.** Number of participants in the study, and excluded participants by various criteria.

same period; these data included investigation made by the Swedish National Food Agency 1988, 1997–1998 and 2010–2011 [4] as well as the annual estimations of the food consumption in Sweden collected by the Swedish Board of Agriculture [23].

The number of participants was considerably higher during 2009–2017 as compared to 2002–2008. Median daily values of energy intake (E% of macronutrients) are presented for the whole period but statistical evaluation of changes in nutrient and food intake was done only on the 2009–2017 cohort. This cohort included 859 individuals, of which 729 (85%) remained after filtering using the 95% Goldberg cutoff, Fig 2.

## Energy intake from protein, fat, carbohydrates, fiber and alcohol 2002–2017

Primary data from each individual was based on self-reported food intake from one weekday and one weekend day, as described above. The average daily intake was calculated for each individual, and descriptive statistics was performed on the group. This included median, mean, standard deviation and quartiles of daily energy intake from protein, fat, carbohydrates, fiber and alcohol expressed as percent of total energy intake (E%) for the period 2002–2017. The Swedish national nutrition database [24] was primarily used in these calculations, but for food items not included in the national database (14% of the items), the United States Department of Agriculture (USDA) database [25] or the producers' own data [26] were used. Standard energy conversion factors were used in the calculations and total energy was calculated as: protein x 17 kJ/g + fat x 37 kJ/g + carbohydrates 17 kJ/g + fiber 8 kJ/g + alcohol 29 kJ/g.

Official statistics of food consumption in Sweden 1980–2016 was extracted from the statistical database available online from the Swedish Board of Agriculture [23] as the mean intake of protein, fat, carbohydrate, fiber and alcohol expressed as kJ/day, which were transformed into E%. These data as well as E% values reported in the national surveys performed by the Swedish National Food Agency in 1988, 1997–1998 and 2010–2011 [4] were used for comparison with the results in the present study, Fig 3. Mean E% values of macronutrients were used in this comparison. As the calculations of energy from fiber has changed between years, the adjusted E% values published in the latest publication from the Swedish National Food Agency [4] was used in the comparison, and energy from fiber was separated from carbohydrates for all years.

## Statistical analysis of nutrient intake and food categories 2009–2017

Reported median daily intake from the years 2009–2017 with regard to fats (saturated fatty acids; SFA, monounsaturated fatty acids; MUFA and polyunsaturated fatty; PUFA), some vitamins (folate and vitamin D) and minerals (iron, potassium and sodium) were calculated using nutritional databases, as described above. Micronutrients included in the study were selected based on the latest national food survey, reporting that their intake contrasted relative to the Nordic nutrition recommendations [4]. Median daily intake of different foods during the same period was calculated by manually assigning all reported foods into either of 25 food categories, designed to correspond to the grouping of the national Swedish survey [4], as described previously [27]. Reported median intake of nutrients, Table 2, and food categories, Table 3, from the 2009–2017 cohort were statistically tested to identify significant changes during the period. Wilcoxon rank sum test was used to determine if values obtained 2009 and 2017 were significantly different. In order to determine that identified differences was due to a gradual change (rather than serendipitous high/low values for these specific years) it was tested if the change between 2009 until 2017 followed a linear trend (i.e., if the slope of a linear

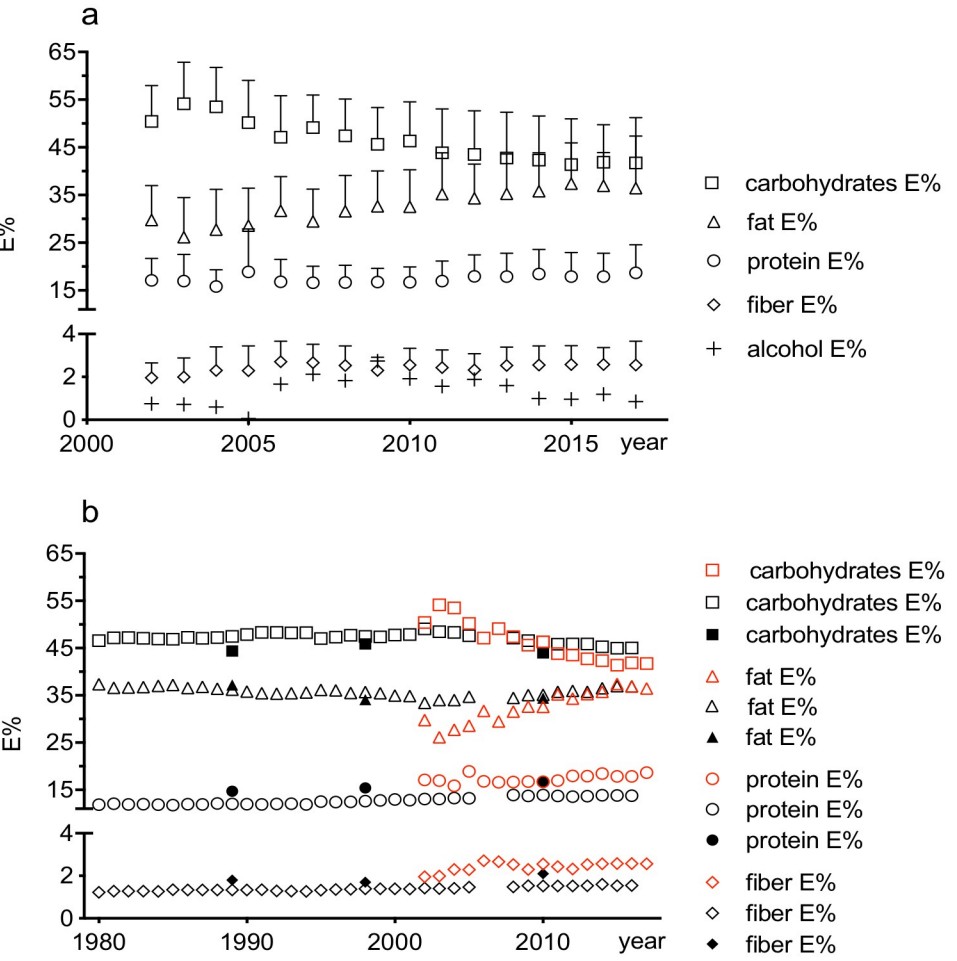

**Fig 3. Energy intake expressed as E% of total energy in the present study and official data from Sweden.** a: Energy intake in the present study (mean E% values with one standard deviation). Alcohol intake varied widely between participants and years and standard deviations are not shown. b: Energy intake (mean E% values) of women in the present study (red open symbols) compared with data from Swedish Board of Agriculture (black, open symbols; including both women and men) and data from the Swedish National Food Agency (black, closed symbols, only women). Alcohol intake was part of the total E% estimation but is not shown.

regression was significantly non-zero). A significant alteration was considered if p-values were equal to, or below 0.01, for both these tests. Statistical analysis was performed with MATLAB 2015a (Mathworks inc, Natick, MA, USA).

## Ethical considerations

The data collection used in the present study was not originally destined for research purposes, but as a teaching and examination component within a university course. Consequently, the students were not asked for consent to use the data in the current analysis. However, the submitted data is of very mildly sensitive nature, and we have made every effort to ensure that there is no way to link the data to individuals. To protect the integrity of the students all data has been anonymized and merged in a single data sheet. In advice from the Swedish Ethical Review Authority ethical permission was not required for publishing the study (Dnr: 2019–03636).

**Table 2. Macronutrients (E%); fiber, fatty acids, vitamins and mineral intake; median daily values.**

| | | 2002 | 2003 | 2004 | 2005 | 2006 | 2007 | 2008 | 2009 | 2010 | 2011 | 2012 | 2013 | 2014 | 2015 | 2016 | 2017 | P1* | P2* |
|---|---|---|---|---|---|---|---|---|---|---|---|---|---|---|---|---|---|---|---|
| MACRONUTRIENTS | Energy (MJ) | 7.91 | 8.28 | 7.99 | 7.73 | 7.90 | 9.20 | 8.83 | 8.85 | 8.69 | 8.85 | 8.31 | 8.54 | 8.14 | 8.88 | 8.90 | 8.24 | p = 0.08 | p = 0.45 |
| | Carb. (E%) | 50.4 | 55.0 | 51.6 | 48.8 | 50.4 | 48.8 | 48.0 | 47.0 | 46.8 | 44.3 | 45.1 | 41.7 | 41.5 | 43.9 | 41.0 | 41.4 | p<0.01 | p<0.01 |
| | Protein (E%) | 15.9 | 16.6 | 14.9 | 16.3 | 15.9 | 16.4 | 16.1 | 16.6 | 16.4 | 16.4 | 17.1 | 17.1 | 17.5 | 17.0 | 16.7 | 18.0 | p = 0.03 | p = 0.03 |
| | Fat (E%) | 30.0 | 24.6 | 28.1 | 29.7 | 30.0 | 30.0 | 31.5 | 31.7 | 31.8 | 34.7 | 34.4 | 34.9 | 35.3 | 36.3 | 36.9 | 37.5 | p<0.01 | p<0.01 |
| | Diet Fib. (E%) | 1.7 | 1.9 | 2.0 | 2.4 | 1.7 | 2.7 | 2.4 | 2.3 | 2.4 | 2.3 | 2.2 | 2.4 | 2.5 | 2.5 | 2.5 | 2.5 | p = 0.05 | p = 0.13 |
| | Alcohol (E%) | 0.0 | 0.0 | 0.0 | 0.0 | 0.0 | 0.0 | 0.0 | 1.5 | 0.0 | 0.0 | 0.0 | 0.0 | 0.0 | 0.0 | 0.0 | 1.8 | n.d. | n.d. |
| | Diet. Fib. g/day | 18.6 | 19.7 | 19.5 | 25.5 | 24.0 | 28.2 | 26.6 | 25.5 | 27.9 | 27.3 | 24.7 | 27.3 | 27.0 | 29.2 | 28.3 | 26.8 | p = 0.09 | p = 0.30 |
| | Diet. Fib. (g/MJ) | 2.18 | 2.33 | 2.54 | 3.02 | 3.06 | 3.32 | 3.00 | 2.74 | 3.18 | 2.90 | 2.75 | 2.97 | 2.91 | 3.00 | 3.04 | 3.06 | p = 0.09 | p = 0.30 |
| FATS | SFA (E%) | n.d. | n.d. | n.d. | n.d. | n.d. | n.d. | n.d | 11.6 | 11.7 | 12.0 | 12.0 | 12.7 | 12.6 | 13.2 | 13.4 | 13.3 | p = 0.12 | p<0.01 |
| | PUFA (E%) | n.d. | n.d. | n.d. | n.d. | n.d. | n.d. | n.d | 4.8 | 5.1 | 5.5 | 5.7 | 5.5 | 6.4 | 6.0 | 6.2 | 7.1 | p<0.01 | p<0.01 |
| | MUFA (E%) | n.d. | n.d. | n.d. | n.d. | n.d. | n.d. | n.d | 12.0 | 12.1 | 13.1 | 12.5 | 13.2 | 13.3 | 13.1 | 14.0 | 14.1 | p<0.01 | p<0.01 |
| VITAMINS & MINERALS MINMINERALS | Vit D (μg/day) | n.d. | n.d. | n.d. | n.d. | n.d. | n.d. | n.d | 4.71 | 4.63 | 4.55 | 5.04 | 6.00 | 6.04 | 6.18 | 6.19 | 6.51 | p = 0.38 | p = 0.01 |
| | Folate (μg/day) | n.d. | n.d. | n.d. | n.d. | n.d. | n.d. | n.d | 313 | 367 | 363 | 294 | 355 | 325 | 384 | 392 | 424 | p<0.01 | p = 0.06 |
| | Fe (mg/day) | n.d. | n.d. | n.d. | n.d. | n.d. | n.d. | n.d | 11.9 | 12.2 | 12.4 | 11.4 | 12.2 | 11.6 | 12.5 | 13.3 | 13.7 | p = 0.45 | p = 0.44 |
| | K (g/day) | n.d. | n.d. | n.d. | n.d. | n.d. | n.d. | n.d | 3.5 | 3.4 | 3.6 | 3.3 | 3.5 | 3.2 | 3.3 | 3.5 | 3.4 | p = 0.13 | p = 0.26 |
| | Na (g/day) | n.d. | n.d. | n.d. | n.d. | n.d. | n.d. | n.d | 2.9 | 2.7 | 2.8 | 2.9 | 2.9 | 2.7 | 3.1 | 2.9 | 2.8 | p = 0.77 | p = 0.38 |

*P1 are the obtained p-values in comparing data from 2009 and 2017, applying the Wilcoxon rank sum test. P2 are the obtained p-values in testing for a linear trend from 2009 to 2017, based on linear regression of the median values. p-values equal to, or below 0.01, are regarded statistically significant. n.d. = not determined. All statistics is used as implemented in MATLAB 2015a.

## Results

### Participants

Food intake data has been collected from 2002 until 2017 from students taking part in a mandatory task that was part of a nutrition course administrated from Linnaeus University, Kalmar, Sweden. The number of drop-outs from the course was higher among women (34%) compared to men (22%), including students who took part in the initial phases of the course, but who did not complete the food survey. Geographically, students from counties close to the Linnaeus University site are over-represented in the study, whereas geographically remote counties are under-represented, Fig 1. A substantial change occurred in 2009 when the course was transferred into a distance format, which affected both the number of participants and their geographical distribution, Fig 1, whereas neither age nor weight changed significantly, Table 1. The average distance between the municipalities of residence of the students and the Linnaeus University site was 175 km 2002–2008, and 347 km 2009–2017.

Standard procedures were used to exclude unrealistic food intake reports using the 95% Goldberg cut-off [22]. The excluded fraction was 17% among women, which agree well with the estimated under-reporters in e.g. the Swedish National Food Agency study 2010–2011 (16% among all women) [4]. The nutrient and food categories of the 2009–2017 cohort was examined in closer detail and on average 81 participants were included in the statistical analysis for each year, which might be comparable to the same age group (18–30 years) in the Swedish National Food Agency study 2010–2011 which consisted of 202 individuals [4]. The average BMI of all women in the Swedish National Food Agency study 2010–2011 was 25 kg/$m^2$ as compared to our study where the average BMI in different years (2009–2017) varied between 22.2–22.8 kg/$m^2$.

**Table 3. Energy intake per meal/food category, median daily intake.**

| | CATEGORY | 2009 | 2010 | 2011 | 2012 | 2013 | 2014 | 2015 | 2016 | 2017 | P1* | P2* |
|---|---|---|---|---|---|---|---|---|---|---|---|---|
| MEALS | Breakfast (E%) | 21.7 | 22.5 | 21.7 | 19.9 | 19.6 | 21.4 | 19.9 | 19.9 | 20.8 | 0.27 | 0.03 |
| | Lunch (E%) | 23.5 | 20.8 | 25.3 | 23.4 | 23.7 | 24.6 | 24.9 | 24.7 | 24.8 | 0.44 | 0.18 |
| | Dinner (E%) | 28.7 | 28.8 | 29.2 | 28.5 | 30.0 | 28.6 | 25.8 | 26.5 | 30.4 | 0.89 | 0.22 |
| | Snack (E%) | 24.2 | 28.1 | 22.7 | 23.9 | 25.5 | 23.9 | 25.0 | 25.0 | 24.0 | 0.72 | 0.12 |
| FOOD CATEGORIES | Spreads & fats (E%) | 1.7 | 2.2 | 3.1 | 2.0 | 3.1 | 2.7 | 2.8 | 2.8 | 5.2 | 0.03 | 0.06 |
| | Cheese (E%) | 3.7 | 2.8 | 4.4 | 3.9 | 4.7 | 5.1 | 5.0 | 4.5 | 5.4 | 0.16 | 0.02 |
| | Milk, Dairy (E%) | 8.0 | 6.9 | 7.4 | 8.2 | 6.5 | 7.1 | 7.6 | 6.7 | 7.5 | 0.01 | 0.19 |
| | Bread (E%) | 7.7 | 10.1 | 9.8 | 8.6 | 8.5 | 7.4 | 7.8 | 7.3 | 8.2 | 0.61 | 0.07 |
| | Potatoes (E%) | 4.8 | 1.8 | 2.9 | 0.0 | 2.2 | 1.6 | 1.1 | 2.5 | 3.9 | 0.04 | 0.43 |
| | Root veg. (E%) | 0.4 | 0.5 | 0.0 | 0.0 | 0.3 | 0.4 | 0.3 | 0.5 | 0.8 | 0.45 | 0.54 |
| | Vegetables (E%) | 4.8 | 3.8 | 4.6 | 2.9 | 5.2 | 5.2 | 5.7 | 6.4 | 6.8 | 0.48 | 0.05 |
| | Fruits, berries (E%) | 6.6 | 7.7 | 6.3 | 6.6 | 7.7 | 6.8 | 7.1 | 8.3 | 7.9 | 0.96 | 0.64 |
| | Fruit juices (E%) | 0.0 | 0.0 | 0.0 | 0.0 | 0.0 | 0.0 | 0.0 | 0.0 | 0.0 | 0.04 | n.d. |
| | Pasta (E%) | 0.0 | 0.0 | 0.0 | 0.0 | 0.0 | 0.0 | 0.0 | 0.0 | 1.8 | 0.22 | n.d. |
| | Meat (E%) | 9.2 | 10.2 | 8.4 | 10.5 | 12.2 | 9.8 | 11.1 | 8.5 | 13.7 | 0.04 | 0.25 |
| | Egg (E%) | 0.5 | 1.5 | 2.4 | 1.8 | 2.0 | 2.5 | 1.4 | 3.8 | 2.8 | <0.01 | 0.06 |
| | Fish (E%) | 3.8 | 4.9 | 2.1 | 2.7 | 3.4 | 4.6 | 3.8 | 5.0 | 4.7 | 0.60 | 0.67 |
| | Sausages (E%) | 0.0 | 0.0 | 0.0 | 0.0 | 0.0 | 0.0 | 0.0 | 0.0 | 1.0 | 0.22 | n.d. |
| | Biscuits (E%) | 5.0 | 0.0 | 0.0 | 2.7 | 1.7 | 0.0 | 0.0 | 0.0 | 4.3 | <0.01 | 0.20 |
| | Ice cream (E%) | 0.0 | 0.0 | 0.0 | 0.0 | 0.0 | 0.0 | 0.0 | 0.0 | 0.7 | 0.52 | n.d. |
| | Pastries (E%) | 0.0 | 0.0 | 0.0 | 0.0 | 0.0 | 0.0 | 0.0 | 0.0 | 0.2 | 0.88 | n.d. |
| | Jams (E%) | 0.0 | 0.0 | 0.0 | 0.0 | 0.0 | 0.0 | 0.0 | 0.0 | 0.5 | 0.13 | n.d. |
| | Soft drinks (E%) | 0.0 | 0.0 | 0.0 | 0.0 | 0.0 | 0.0 | 0.0 | 0.0 | 0.4 | 0.45 | n.d. |
| | Sweets (E%) | 1.9 | 4.0 | 0.8 | 0.0 | 2.4 | 1.2 | 2.3 | 3.1 | 4.2 | 0.23 | 0.75 |
| | Sugar (E%) | 0.0 | 0.0 | 0.0 | 0.0 | 0.0 | 0.0 | 0.0 | 0.0 | 0.3 | 0.03 | n.d. |
| | Coffee & tea (E%) | 0.3 | 0.2 | 0.2 | 0.3 | 0.1 | 0.1 | 0.2 | 0.2 | 0.2 | 0.02 | 0.11 |
| | Alcoholic bev (E%) | 2.3 | 0.0 | 0.0 | 0.0 | 0.0 | 0.0 | 0.0 | 0.0 | 0.9 | <0.01 | 0.13 |
| | Nuts & grains (E%) | 2.1 | 4.7 | 4.2 | 3.5 | 3.2 | 4.1 | 6.7 | 8.0 | 6.2 | 0.02 | 0.03 |
| | Cereals (E%) | 10.0 | 10.1 | 8.9 | 9.8 | 8.4 | 10.8 | 9.7 | 12.6 | 11.8 | 0.13 | 0.24 |

*P1 are the obtained p-values in comparing data from 2009 and 2017, applying the Wilcoxon rank sum test. P2 are the obtained p-values in testing for a linear trend from 2009 to 2017, based on linear regression of the median values. p-values equal to or below 0.01 is regarded statistically significant. n.d. = Not determined. All statistics is used as implemented in MATLAB 2015a.

## Macronutrient intake

The food intake among female students between the years 2002–2017 subdivided into macronutrients expressed as energy percent (mean E%) is shown in Fig 3a and Table 2, and presented together with official data in Fig 3b.

The overall trend in our data indicates that the carbohydrate proportion decreased consistently throughout the sampling period (2002–2017), paralleled by increased fat and protein energy percent. The change from 2009–2017 was evaluated in closer detail, and during this period the carbohydrate intake (median E%) decreased from 45.6% to 41.7%, while fat intake (median E%) increased from 32.6% to 36.4%. The change between 2009 and 2017 is statistically significant for carbohydrates and fat (p<0.01), and show a gradual shift over the period consistent with a linear slope (p<0.01), Table 2.

The protein intake (median E%) during the period increased from 16.6 E% to 18.0 E% but the change is less evident in the statistical evaluation (p = 0.03 for both tests), Table 2. Fiber

intake was stable throughout the period and close to the recommended level of 3 g/MJ, in contrast to the national survey where young women (18–30 years) reported a fiber intake of 2.3 g/MJ [4]. The intake of saturated as well as unsaturated fatty acids (SFA, PUFA and MUFA) increased in terms of total amount, due to the increased fat intake. The increase in intake of PUFA and MUFA was statistically significant (p<0.01 in both tests) but the increase in SFA was not statistically significant (p = 0.12) although a significant linear trend was seen (p<0.01), Table 2. The energy intake (mean E%) available from official reports in Sweden are in line with the obtained results, although our data indicate a steeper trend, Fig 3b.

## Micronutrient intake

The calculated intake of a few selected vitamins and minerals was evaluated in the 2002–2017 cohort in order to detect if the change in energy composition might affect micronutrient intake, Table 2. No changes in micronutrient intake fulfilled both criteria for a statistically significant alteration. The vitamin D intake varied between 4.17 μg/day (2009) to 6.51 μg/day (2017) as compared to the recommendation of 10 μg/day (NNR 2012, [6]). The difference between 2002 and 2017 was not statistically significant (p = 0.38), albeit that a positive linear trend was observed over the time interval (p = 0.01). Folate intake for the 2009–2017 cohort varied between 313 μg/day (2009) and 424 μg/day (2017), which is substantially higher than the national average for females 18–30 years (223 μg/day; [4]) and close to the recommended level of 400 μg/day [6]. The difference between 2009 and 2017 is statistically significant (p<0.01) but the linear trend is not (p = 0.06). Iron intake (11.4–13.7 mg/day, Table 2) is low relative to recommendations (15 mg/day [6]) but high relative to the national average (females: 18–30 years 8.9 mg/day [4]). Sodium intake did not appear to deviate from the national average, albeit high relative to NNR 2012 recommendations (2.7–3.1 g/day as compared to recommendations <2.4 g/day). Potassium intake (3.2–3.6 g/day) was higher than the national average (females: 18–30 years 2.7 g/day; [4]) as well as recommendations (3.1 g/day [6]). No statistical analysis was performed to compare the results of the present study with official values, as the standard deviation is unknown for the latter.

## Food categories

To investigate if the observed changes in energy composition over time could be explained by differences in food category choices, the change in reported intake within these categories were subject to statistical analysis, Table 3. No food category fulfilled both criteria for a statistically significant change, but three food categories show a distinctive increase: "spreads and fats" (p = 0.03), "egg" (p<0.01) and "nuts and grains" (p = 0.02).

## Discussion

Official reports [4, 7] as well as scientific publications [1, 2] have indicated that a change in diet of the Swedish population is ongoing, and concerns about the health effects of a diet with lower carbohydrate and higher fat content have been raised [28]. Information about the development of this large-scale change in the Swedish diet is therefore of importance. Our study covers the time period of interest (2002–2017) and it is a sample from a relatively young (mean age 30 years), lean (average BMI < 23) and well-educated population from the south-east of Sweden (Table 1 and Fig 1). The change in food intake composition expressed as E% of this group does however reflect the overall national trend, as seen in Fig 3b, with a clear shift towards lower energy intake from carbohydrates in combination with higher energy intake from fat and proteins, expressed as E% of each nutrient. Changes in carbohydrate and fat intake are statistically significant in our study, while the increase in protein intake is smaller

and not statistically significant (p = 0.03). The observed change in diet in our study group seems to coincide with a higher intake of foods such as spreads, fats, egg, nuts and grains; although not statistical evident. A correct estimation of nutrient intake is dependent on the reliability of the food data base and food categories used [29]. National food databases are continuously updated, but it is difficult to cover all new food items. The Swedish nutrition database was mainly used in the present study, but a few items were collected from other sources. Another limitation is that whole grain products are combined with refined products when food categories are made, making it impossible to evaluate the fiber intake in closer detail.

We believe that the raised consumer-interest in low-carb, high-fat diets [9, 10] can be linked to our results. During this period, negative attitudes towards fat intake have been receding, as reflected by a recent survey of Scandinavians' attitudes to various food components [11]. While the surge of fad diets has been interpreted as a distrust of science-based recommendations [8], the trend towards higher fat and lower carbohydrate intake actually represents a continuation of a trend that has been ongoing since the 1940's [30] and the increased fat consumption in recent years is a return to the level present in Sweden prior to 1980 [1]. It is however striking that carbohydrate intake is at a historical low, and that it does not adhere to the Nordic nutrition recommendations [6]. It is noteworthy that a similar trend of increasing fat and decreasing carbohydrate composition could be seen when studying variation in macronutrient composition in recipes during 1970–2010 [31].

Some caution is required in the interpretation of our results. This investigation was not initially designed as a research investigation, but as a mandatory annually reoccurring task in a university course segment, and it was much later, in the face of the amount of data collected, that we decided to analyze potential trends over time. It is not surprising then to find that some aspects of data collection ideally would have been planned and performed differently. As stated, the proportion of men participating in the course on an annual basis was very low and we thus selected to exclude this data from the manuscript (although the trend appears to be similar for men as for women). The study group sample is also limited in its geographical distribution, following a trend (although not without exceptions), where counties in the vicinity of Linnaeus University (Kalmar and Kronoberg counties) are over-represented, and counties further away are under-represented, see Fig 1. Furthermore, the motivation of study participants to maintain their usual eating habits and perform registration truthfully may potentially have been affected by the mandatory format. Background factors such as weight and length, are self-reported and not standardized. On the other hand, we perceive that the course, not being mandatory itself, is likely to attract students with stronger interest and more knowledge in food and nutrition than the general population. The proportion of drop-outs from the course was 34% among women. The reasons for dropping out have not been investigated, but it is not a far-fetched assumption that the remaining students (66%) were more strongly motivated to carry out the food recording and to finalize the course. We do not know whether the sample represents physically active students or not, and the PAL level used is arbitrary in order to allow the application of cut-off calculations. In addition to the known limitations in reliability imposed by self-reporting [32, 33], generalizability is also limited by the fact that data was collected during a limited period each spring, not allowing to take into account intra-annual variations. Moreover, only two days of data reporting from each student were included (one weekday and one weekend-day) which is probable to have an effect on the absolute values, although most likely consistently over the time-period.

Despite these limitations, there is little to threaten the overall conclusion that the energetic contributions from protein, fat and carbohydrates have changed consistently over a long period of time for this group of nutrition-interested students. However, we do not suggest that this trend is necessarily unfavorable. While it may be somewhat worrying that the

carbohydrate (E%) intake is presently below the NNR 2012 recommendations, fiber intake is relatively high in comparison to national data. This shows that a low carbohydrate intake does not have to coincide with a low fiber intake. Furthermore, fat intake (E%) is close to the upper limit stated in NNR 2012 but it does not appear to have altered the relative proportions of fatty acids, *e.g.* SFA, MUFA and PUFA. The increase in SFA is however not congruent with dietary advice from the Swedish National Food Agency (6). The intake of commonly limiting health-related micronutrients such as vitamin D and folate is increasing. Moreover, folate, iron and potassium intake are high upon a national comparison. The question now remains to what extent these students differ from the general public in terms of food intake. The students' intake follows the general trend of the Swedish population in terms of energy composition but the slope appears steeper. This could indicate that this sample is more sensitive to societal food trends than the population at large. Further investigation of this group is required, and indeed ongoing, to ascertain if this is the case. It is well-known that correlations exist between food choices and *i.e.* attitudes and interests [34], cultural background [35, 36], physical activity [37], self-esteem and self-image [38], socio-economic status [39], and subject knowledge [40, 41], but such sub-division is not routinely employed in nutritional surveys.

## Conclusions

Our results suggest that the observed national transition from carbohydrate to fat intake persists in Sweden, and that it might be especially evident among individuals interested in food and nutrition. Carbohydrate intake is at present lower than recommended by the Swedish National Food Agency, and fat intake is at the upper limit. We argue that our study, indicating a similar but steeper dietary shift than in the population as a whole, suggests that monitoring the dietary intake of subgroups can be a useful strategy to extrapolate potential effects of ongoing trends.

## Supporting information

**S1 Data. 2002–2017.**
(XLSX)

## Acknowledgments

The authors express their gratitude to Mrs Sara Boman for analysis of part of this dataset in a previous bachelor project, and to Mr Björn Magnusson for assistance in data accession.

## Author Contributions

**Conceptualization:** Håkan S. Andersson.

**Data curation:** Anna Blücher.

**Formal analysis:** Maria Bergström, Andreas Håkansson.

**Visualization:** Maria Bergström, Håkan S. Andersson.

**Writing – original draft:** Maria Bergström, Håkan S. Andersson.

**Writing – review & editing:** Maria Bergström, Andreas Håkansson, Anna Blücher, Håkan S. Andersson.

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
