## [Decision Letter · Decision Letter 0]

11 Nov 2019

PONE-D-19-24477

From carbohydrates to fat: Trends in food intake among Swedish nutrition students from 2002 to 2017

PLOS ONE

Dear Dr Bergström,

Thank you for submitting your manuscript to PLOS ONE. After careful consideration, we feel that it has merit but does not fully meet PLOS ONE’s publication criteria as it currently stands. Therefore, we invite you to submit a revised version of the manuscript that addresses the points raised during the review process.

Specifically, major concerns are pointed for methodological description flaws present in the manuscript. I would like to stress the points that reviewer #1 has found. More specifically, the context of your results in light of already available similar studies have been pointed out, and need to be solved.

We would appreciate receiving your revised manuscript by Dec 26 2019 11:59PM. To enhance the reproducibility of your results, we recommend that if applicable you deposit your laboratory protocols in protocols.io, where a protocol can be assigned its own identifier (DOI) such that it can be cited independently in the future. For instructions see: http://journals.plos.org/plosone/s/submission-guidelines#loc-laboratory-protocols

We look forward to receiving your revised manuscript.

Kind regards,

Jose M. Moran

Academic Editor

PLOS ONE

Journal Requirements:

2. Please include captions for your Supporting Information files at the end of your manuscript, and update any in-text citations to match accordingly. Please see our Supporting Information guidelines for more information: http://journals.plos.org/plosone/s/supporting-information

Reviewers' comments:

Reviewer's Responses to Questions

**Comments to the Author**

1. Is the manuscript technically sound, and do the data support the conclusions?

Reviewer #1: Yes

Reviewer #2: Yes

2. Has the statistical analysis been performed appropriately and rigorously? 

Reviewer #1: No

Reviewer #2: Yes

3. Have the authors made all data underlying the findings in their manuscript fully available?

Reviewer #1: Yes

Reviewer #2: Yes

4. Is the manuscript presented in an intelligible fashion and written in standard English?

Reviewer #1: Yes

Reviewer #2: Yes

5. Review Comments to the Author

Reviewer #1: The paper by Bergström and colleagues provides insights into dietary trends over an 8 year period in Sweden among female nutrition students. While the sampling of this population is not representative of the Swedish population, the study is unique in that it tracks a subgroup of the Swedish population over time. The authors have done a nice job of placing these findings into the broader nutritional/dietary context in Sweden. While the results are interesting, their presentation could be strengthened through the inclusion of additional methodological details. The study could also benefit from further details on the type of carbohydrates that are being consumed over time such as refined grains versus whole grains.

Specific Comments:

1. The title, abstract (line 24) and discussion (line 319-320) highlight that study time period as being 2002 to 2017; however, the study focuses the analysis on the time period of 2009 to 2017. These sections of the manuscript should be changed to reflect that the main analysis is done using data from 2009 and 2017 and the methodology section can include the details on the full study period.

2. Line 40: change “begun” to “began”

3. The study that is referenced in the introduction from lines 74 to 83 was described at line 74 as a “25-year long cross-sectional study.” Was this a series of cross-sectional studies conducted at different time points within that time frame, a rolling study that continuously collected information on participants, or were participants tracked over time? Additional clarity on the type of study being described would be helpful.

• Further, at line 77, it states that “…in the beginning of the intervention program…” Was there an interventional aspect to this study? Again, more precision on the design of the study being described would be helpful to the reader.

4. I’d suggest editing the sentence at line 112 to the following “The association of overweight and obesity with chronic diseases, e.g. type 2 diabetes, high blood pressure, cardiovascular disease, asthma and arthritis, is well established. Diet can affect the development of chronic diseases, either directly or mediated through the development of overweight and obesity, and thus warrants close monitoring in the general population.”

5. It should be clearly stated in the study design section that this was a series of cross-sectional studies.

6. The figures at the end of the manuscript are not labeled with their numbers.

7. Lien 146: Are the results the average of each individuals average 2 day intake? This should be stated in the methods.

8. Line 154: Include a sensitivity analysis with the male participants. Is there any reason to believe that their dietary results would be so different that it would obscure the trends seen in women?

9. Line 157: The characteristics of the excluded participants should be provided along with some information as to whether most of those excluded under- or over-reported calorie intake relative to requirements. Also, since you are just reporting the mean intake, rather than doing any type of regression analysis, you could also include all of the values and report the median, which will not be influenced by extreme outliers.

10. Table 1: The line that only includes the value of 729 is not easy to understand. I’d exclude this line from table 1.

11. Lines 188 to 191: more explanation on how the percent of energy from fiber was calculated. What value in kcal/g was used as the energy density for fiber? Was the same value used for all types of fiber i.e. soluble and insoluble? Was 4kcal/g used as the energy density of carbohydrates and protein and 9 kcal/g for fat? This information should also be included in the methodology section.

12. Was the height and weight information in Table 1 self-reported? How this information was collected should be included in the methodology section and this limitation should be noted in the discussion section.

13. Line 207: change “follow” to “followed”

14. Figure 2: More methodological details on the data from the Swedish Board of Agriculture and the Swedish National Food Agency should be included in the methods section.

15. Table 2: Report the p value rather than just “<0.01”

16. Line 279: It was noted that there was not a statistically significant difference in SFA intake over the time period but the authors didn’t note that there was a statistically significant trend. This should be included in the results.

17. The rationale for reporting only vitamin D, folate, iron, potassium and sodium, and not other nutrients should be included in the methods section.

18. Starting after page 8, the page number does not continue sequentially.

19. Line 288: your statistical test is to determine if there is greater variability between the years rather than within each year – it doesn’t test specifically to see if there was an increase during the time period.

20. Lines 283 to 300: Note in the description of the results in this section that comparisons to national averages were descriptive only and statistical analyses weren’t done.

21. Table 3: include the SD or SE in the table for each year.

22. Line 356: it was noted that reporting food intake was a mandatory component of the course yet the response rate was still only 66% for women? Was this due to lost data or to the participants not submitting the assignment? Additional explanation would be helpful in the discussion.

23. The main findings of the study are that carbohydrate intake has decreased over time among this group. It would be helpful to understand how refined grain and whole grain intake also changed during this time period. Many national dietary guidelines recommend the consumption of whole grains; therefore, it would be helpful to understand if the proportion of refined grains to whole grains changed during this period.

24. Line 376: It should be noted in this paragraph that the percentage of energy from saturated fat increased which is not congruent with most authoritative dietary guidances.

Reviewer #2: This research is of interest . The paper needs of some suggestions :

In Introduction the authors should underline the linkahe between Food consumption data and Food Composition data and adding related references such as: Durazzo et al. Food Groups and Individual Foods: Nutritional Attributes and Dietary Importance.In book: Reference Module in Food Science

DOI: 10.1016/B978-0-08-100596-5.21337-1.

In Material and Methods a scheme of design of study should be added. In subparagraph "Energy intake from protein, fat, carbohydrates, fiber and alcohol 2002-2017", at lines 179-181 the authors should underline the importance of a common language of database in order to allow an exchange/ a comparison of data as well as the study of dish preparation in national database. In this regard proper references should be added such as :Durazzo et al. 2019 Nutritional composition and dietary intake of composite dishes traditionally consumed in Italy. Journal of Food Composition and Analysis.Journal of Food Composition and Analysis 77 DOI: 10.1016/j.jfca.2019.01.007; Durazzo et al. 2019; Italian composite dishes: description and classification by LanguaL™ and FoodEx2. 2019 European Food Research and Technology; Marconi et al. 2018. Food Composition Databases: Considerations about Complex Food Matrices.Foods. 2018 7(1). pii: E2. doi: 10.3390/foods7010002. In subparagraph "Statistical analysis of nutrient intake and food categories" The authors should mention the importance of choice of food groups and variability along databases.

A subparagraph Conclusion should be added.

6. PLOS authors have the option to publish the peer review history of their article (what does this mean?). If published, this will include your full peer review and any attached files.

Reviewer #1: Yes: Jessica Smith

Reviewer #2: No

---

## [Author Response · Author response to Decision Letter 0]

19 Dec 2019

We are grateful for constructive comments and suggestions for improvements regarding our manuscript. We believe that we have been able to meet all remarks from reviewers and will explain our actions below.

Response to Reviewer #1.

Comment 1: We agree with the reviewer that the manuscript benefits from an improved description of what data that was statistically evaluated. We have added one figure (as suggested by Reviewer 2) and changed the phrasing in the abstract to clarify that data from 2009-2017 was in focus for the statistical evaluation. We do however believe that it is important to present data from the beginning of the period (2002-2008) to allow comparisons of the long-time changes evident in our data to those of official sources, as presented in Figure 3 in the revised manuscript.

Comment 2: Revised manuscript is modified according to Reviewer´s suggestion.

Comment 3: Revised manuscript has been rephrased to reflect that the Northern Sweden Diet Database (NSDD), which is part of the Västerbotten intervention program (VIP) and the Northern Sweden MONICA project, are large ongoing independent cross-sectional health surveys. The data that we use was not part of any intervention program and line 77 has been changed.

Comment 4: Revised manuscript is modified according to Reviewer´s suggestion.

Comment 5: Revised manuscript is modified according to Reviewer´s suggestion.

Comment 6: It is our understanding that figure number should not be included in the figure; only in the figure legend.

Comment 7: Revised manuscript is modified to clarify that it is the average daily intake for each individual (mean value of two reported days) that is used in all calculations.

Comment 8: It is difficult to say to what extent the intake of male consumers with the same demographic characteristics would differ from the female ones. Unfortunately, the number of male participants was too low to allow a meaningful comparison between female and male consumers. This is a limitation of the study, and we have clarified in the introduction that only women were included. This is also evident in the new figure added to the manuscript.

Comment 9: Almost all excluded values were due to underreporting, similar to what has been seen in previous investigations and the revised manuscript has been modified to reflect this fact. A median value, as an alternative to Goldberg cut-off, is likely to skew the results as the number of participants eating below cut off by far exceeds the number eating beyond cut off. 

Comment 10: Revised manuscript is changed according to Reviewer´s suggestion.

Comment 11: Standard energy conversion factors were used in the calculations and total energy was calculated as: protein x 17 kJ/g + fat x 37 kJ/g + carbohydrates 17 kJ/g + fiber 8 kJ/g + alcohol 29 kJ/g. This information has been included in the methodology section. Soluble and insoluble fiber were not identified; only total fiber. 

Comment 12: Height and weight were self-reported by the participants. This has been included in the methodology section and discussion in the revised manuscript.

Comment 13: Revised manuscript is changed according to Reviewer´s suggestion.

Comment 14: Official statistics of food consumption in Sweden 1980-2016 was extracted from the statistical database available online from the Swedish Board of Agriculture as the mean intake of protein, fat, carbohydrate, fiber and alcohol expressed as kJ/day, which were transformed into E%. These data as well as E% values reported in the national surveys performed by the Swedish National Food Agency in 1988, 1997-1998 and 2010-2011 were used for comparison with the results in the present study. The manuscript has been changed to reflect this.

Comment 15: Revised manuscript is changed according to Reviewer´s suggestion.

Comment 16: Revised manuscript is changed according to Reviewer´s suggestion.

Comment 17: Micronutrients included in the study were selected based on the latest national (Swedish) food survey, reporting that their intake contrasted relative to the Nordic nutrition recommendations. This is clearly stated in the revised manuscript.

Comment 18: Page numbers have been corrected.

Comment 19: We have reformulated the text to clarify that the difference between 2002 and 2017 in vitamin D intake is not statistically significant (p=0.38), but a positive linear trend was observed over the time interval (p=0.01).

Comment 20: Revised manuscript is changed according to Reviewer´s suggestion.

Comment 21: Values in table 3 do not have a normal distribution and median values were used in the statistical evaluation (not mean values). Standard deviation has not been calculated for these values as the Wilcoxon rank sum test was used in statistical analysis.

Comment 22: The number of drop-outs from the course was 34% among women, which involved students who took part in the initial phases of the course, but who did not complete the food survey. We have rephrased the text in the methodology section and included a remark in the discussion to address this fact.

Comment 23: We agree with the reviewer that it would be most interesting to know more about the intake of whole grains in our study. We have used the standardized food categories used by the National Food Agency in Sweden, and these do not differentiate between whole grain products and products based on refined flour. We can however see in our data that intake of nuts and grains has a distinctive increase, which we also discuss. We have added a paragraph to discuss the limitation of food categories and also the need of improving food databases in this respect.

Comment 24: Revised manuscript is changed according to Reviewer´s suggestion.

Response to reviewer #2

1. The importance of food groups and variability among databases are discussed, and the work of Marconi, S., Durazzo, A., and Camilli, E. et.al is acknowledged.

2. A scheme of design of the study has been added to the revised manuscript.

3. A conclusion has been added to the revised manuscript as suggested by the reviewer.

---

## [Decision Letter · Decision Letter 1]

10 Jan 2020

From carbohydrates to fat: Trends in food intake among Swedish nutrition students from 2002 to 2017

PONE-D-19-24477R1

Dear Dr. Bergström,

We are pleased to inform you that your manuscript has been judged scientifically suitable for publication and will be formally accepted for publication once it complies with all outstanding technical requirements.

With kind regards,

Jose M. Moran

Academic Editor

PLOS ONE

Additional Editor Comments (optional):

Reviewers' comments:

Reviewer's Responses to Questions

**Comments to the Author**

1. If the authors have adequately addressed your comments raised in a previous round of review and you feel that this manuscript is now acceptable for publication, you may indicate that here to bypass the “Comments to the Author” section, enter your conflict of interest statement in the “Confidential to Editor” section, and submit your "Accept" recommendation.

Reviewer #1: All comments have been addressed

Reviewer #2: (No Response)

2. Is the manuscript technically sound, and do the data support the conclusions?

Reviewer #1: Yes

Reviewer #2: (No Response)

3. Has the statistical analysis been performed appropriately and rigorously? 

Reviewer #1: Yes

Reviewer #2: (No Response)

4. Have the authors made all data underlying the findings in their manuscript fully available?

Reviewer #1: No

Reviewer #2: (No Response)

5. Is the manuscript presented in an intelligible fashion and written in standard English?

Reviewer #1: Yes

Reviewer #2: (No Response)

6. Review Comments to the Author

Reviewer #1: (No Response)

Reviewer #2: (No Response)

7. PLOS authors have the option to publish the peer review history of their article (what does this mean?). If published, this will include your full peer review and any attached files.

Reviewer #1: No

Reviewer #2: No

---

## [Editor Report · Acceptance letter]

14 Jan 2020

PONE-D-19-24477R1 

From carbohydrates to fat: Trends in food intake among Swedish nutrition students from 2002 to 2017 

Dear Dr. Bergström:

I am pleased to inform you that your manuscript has been deemed suitable for publication in PLOS ONE. Congratulations! Your manuscript is now with our production department. 

With kind regards,

on behalf of

Dr. Jose M. Moran 

Academic Editor

PLOS ONE